# Morpho-MNIST: Quantitative Assessment and Diagnostics for Representation Learning

## Abstract

Revealing latent structure in data is an active field of research, having introduced exciting technologies such as variational autoencoders and adversarial networks, and is essential to push machine learning towards unsupervised knowledge discovery. However, a major challenge is the lack of suitable benchmarks for an objective and quantitative evaluation of learned representations. To address this issue we introduce Morpho-MNIST, a framework that aims to answer: "to what extent has my model learned to represent specific factors of variation in the data?" We extend the popular MNIST dataset by adding a morphometric analysis enabling quantitative comparison of trained models, identification of the roles of latent variables, and characterisation of sample diversity. We further propose a set of quantifiable perturbations to assess the performance of unsupervised and supervised methods on challenging tasks such as outlier detection and domain adaptation.

## 1 Introduction

A key factor for progress in machine learning has been the availability of well curated, easy-to-use, standardised and sufficiently large annotated datasets for benchmarking different algorithms and models. This has led to major advances in speech recognition, computer vision, and natural language processing. A commonality between these tasks is their natural formulation as supervised learning tasks, wherein performance can be measured in terms of accuracy on a test set.

The general problem of representation learning (i.e. to reveal latent structure in data) is more difficult to assess due the lack of suitable benchmarks. Although the field is very active, with many recently proposed techniques such as probabilistic autoencoders and adversarial learning, it is less clear where the field stands in terms of progress or which approaches are more expressive for specific tasks. The lack of reproducible ways to quantify performance has led to subjective means of evaluation: visualisation techniques have been used to show low-dimensional projections of the latent space and visual inspection of generated or reconstructed samples are popular to provide subjective measures of descriptiveness. On the other hand, the quality of sampled images generally tells us little about how well the learned representations capture known factors of variation in the training distribution. In order to advance progress, the availability of tools for objective assessment of representation learning methods seems essential yet lacking.

This paper introduces Morpho-MNIST, a collection of shape metrics and perturbations, in a step towards quantitative assessment of representation learning. We build upon one of the most popular machine learning benchmarks, MNIST, which despite its shortcomings remains widely used. While MNIST was originally constructed to facilitate research in image classification, in the form of recognising handwritten digits (LeCun et al., 1998), it has found its use in representation learning, for example, to demonstrate that the learned latent space yields clusters consistent with digit labels. Methods aiming to disentangle the latent space claim success if individual latent variables capture specific style variations (e.g. stroke thickness, sidewards leaning digits and other visual characteristics).

The main appeal of selecting MNIST as a benchmark for representation learning is that, while manifesting complex interactions between pixel intensities and underlying shapes, it has well understood and easily measurable factors of variation. More generally, MNIST remains popular in practice due to several factors: it allows reproducible comparisons with previous results reported in the literature; the dataset is sufficiently large for its complexity and consists of small, two-dimensional greyscale images defining a tractable ten-class classification problem; computation and memory requirements

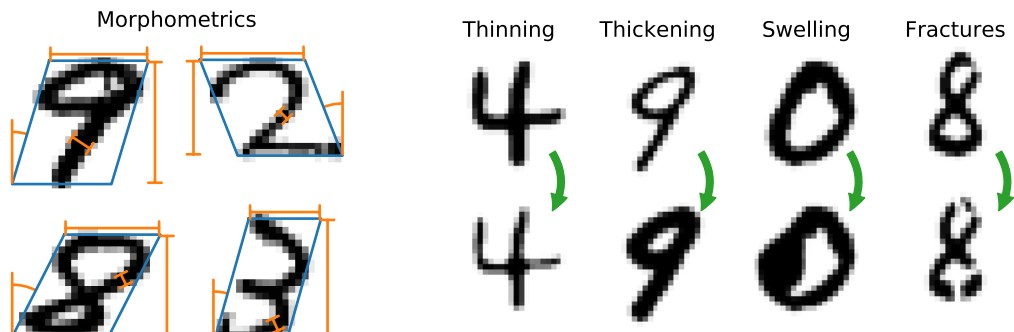

Figure 1: *Left:* MNIST morphometrics—stroke thickness and length (not shown), width, height and slant of digits. *Right:* MNIST perturbations (many more examples of each type in Appendix B).

are low; most popular deep learning frameworks and libraries offer tutorials using MNIST, which makes it straightforward for new researchers to enter the field and to experiment with new ideas and explore latest developments. We take advantage of these qualities and extend MNIST in multiple ways, as summarised in the following.

## 1.1 CONTRIBUTIONS

Our aim is to bridge the gap between methodology-focused research and critical real-world applications that could benefit from latest machine learning methods. As we preserve the general properties of MNIST—such as image size, file format, numbers of training and test images, and the original ten-class classification problem—we believe this new quantitative framework for assessing representation learning will experience widespread use in the community and may inspire further extensions facilitated by a publicly available Morpho-MNIST code base.

**Morphometrics:**  We propose to describe true and generated digit images in terms of measurable shape attributes. These include stroke thickness and length, and the width, height, and slant of digits (cf. Fig. 1, left). Whereas some of these properties have been analysed qualitatively in previous work, we demonstrate that objectively quantifying each of them allows to identify the role of inferred representations. Moreover, these tools can be used to measure model samples, enabling assessment of *generative* performance with respect to sample diversity (Section 4.1) and disentanglement of latent variables (Section 4.2).

These measurements can be directly employed to re-evaluate existing models and may be added retrospectively to previous experiments involving the original MNIST dataset. Adoption of our morphometric analysis may provide new insights into the effectiveness of representation learning methods in terms of revealing meaningful latent structures. Furthermore, for other datasets it suffices to design the relevant scalar metrics and include them in the very same evaluation framework.

**Perturbations:**  We introduce a set of parametrisable global and local perturbations, inspired by natural and pathological variability in medical images. Global changes involve overall thinning and thickening of digits, while local changes include both swelling and fractures (see examples on the right in Fig. 1 and many more in Appendix B). Injecting these perturbations into the dataset adds a new type of complexity to the data manifold and opens up a variety of interesting applications.

The proposed perturbations are designed to enable a wide range of new studies and applications for both supervised and unsupervised tasks. Detection of 'abnormalities' (i.e. local perturbations) is an evident application, although more challenging tasks can also be defined, such as classification from noisy/corrupted data, domain adaptation, localisation of perturbations, characterising semantics of learned latent representations, and more. We explore a few supplementary examples of supervised tasks in Appendix D.

## 1.2 RELATED WORK: DATASETS

In this section, we provide an overview of some datasets that are related to MNIST, by either sharing its original source content, containing transformations of the original MNIST images or being distributed in the same format for easy replacement. We also mention a few prevalent datasets of images with generative factor annotations, similarly to the morphometrics proposed in this paper.

**NIST datasets:** The MNIST (modified NIST) dataset (LeCun et al., 1998) was constructed from handwritten digits in NIST Special Databases 1 and 3, now released as Special Database 19 (Grother and Hanaoka, 2016). Cohen et al. (2017) generated a much larger dataset based on the same NIST database, containing additional upper- and lower-case letters, called EMNIST (extended MNIST).

**MNIST perturbations:** The seminal paper by LeCun et al. (1998) employed data augmentation using planar affine transformations including translation, scaling, squeezing, and shearing. Loosli et al. (2007) employed random elastic deformations to construct the Infinite MNIST dataset. Other MNIST variations include rotations and insertion of random and structured background (Larochelle et al., 2007), and Tieleman (2013) applied spatial affine transformations and provided ground-truth transformation parameters.

**MNIST format:** Due to the ubiquity of the MNIST dataset in machine learning research and the resulting multitude of compatible model architectures available, it is appealing to release new datasets in the same format ($28 \times 28$, 8-bit grayscale images). One such effort is Fashion-MNIST (Xiao et al., 2017), containing images of clothing articles from ten distinct classes, adapted from an online shopping catalogue. Another example is notMNIST (Bulatov, 2011), a dataset of character glyphs for letters 'A'–'J' (also ten classes), in a challengingly diverse collection of typefaces.

**Annotated datasets:** Computer vision datasets that are popular for evaluating disentanglement of learned latent factors of variation include those from Paysan et al. (2009) and Aubry et al. (2014). They contain 2D renderings of 3D faces and chairs, respectively, with ground-truth pose parameters (azimuth, elevation) and lighting conditions (faces only). A further initiative in that direction is the *dSprites* dataset (Matthey et al., 2017), which consists of binary images containing three types of shapes with varying location, orientation and size. The availability of the ground-truth values of such attributes has motivated the accelerated adoption of these datasets in the evaluation of representation learning algorithms.

## 1.3 RELATED WORK: QUANTITATIVE EVALUATION

Evaluation of representation learning is a notoriously challenging task and remains an open research problem. Numerous solutions have been proposed, with many of the earlier ones focusing on the test log-likelihood under the model (Kingma and Welling, 2013) or, for likelihood-free models, under a kernel density estimate (KDE) of generated samples (Goodfellow et al., 2014; Makhzani et al., 2015)—being shown not to be reliable proxies for the true model likelihood (Theis et al., 2016).

Another perspective for evaluation of generative models of images is the visual fidelity of its samples to the training data, which would normally require manual inspection. To address this issue, a successful family of metrics have been proposed, based on visual features extracted by the Inception network (Szegedy et al., 2016). The original Inception score (Salimans et al., 2016) relies on the 'crispness' of class predictions, whereas the Fréchet Inception distance (FID) (Heusel et al., 2017) and the kernel Inception distance (KID) (Bińkowski et al., 2018) statistically compare high-level representations instead of the final network outputs.

Although the approaches above can reveal vague signs of mode collapse, it may be useful to diagnose this phenomenon on its own. With this objective, Arora et al. (2018) proposed to estimate the support of the learned distribution (assumed discrete) using the birthday paradox test, by counting pairs of visual duplicates among model samples. Unfortunately, the adoption of this technique is hindered by its reliance on manual visual inspection to flag identical images.

There have been several attempts at quantifying representation disentanglement performance. For example, Higgins et al. (2017) proposed to use the accuracy of a simple classifier trained to predict

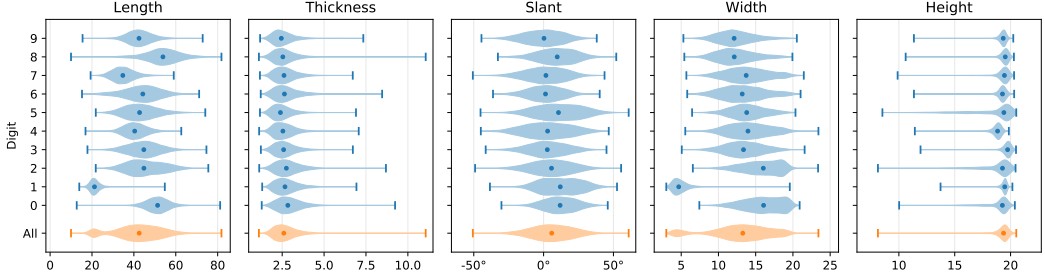

Figure 2: Stages of the image processing pipeline. *Left to right:* original image, upscaled image, binarised image, distance transform, skeleton, downscaled image.

which factor of variation was held fixed in a simulated dataset. There exist further information-theoretic approaches, involving the KL divergence contribution from each latent dimension (Dupont, 2018) or their mutual information with each known generative factor (Chen et al., 2018). Yet another method, explored in Kumar et al. (2018), is based on the predictive accuracy of each latent variable to each generative factor (continuous or discrete).

## 2 MORPHOMETRY

Meaningful morphometrics are instrumental in characterising distributions of rasterised shapes, such as MNIST digits, and can be useful as additional data for downstream learning tasks. We begin this section by describing the image processing pipeline employed for extracting the metrics and for applying perturbations (Section 3), followed by details on the computation of each measurement.

### 2.1 PROCESSING PIPELINE

The original 28×28 resolution of the MNIST images is generally not high enough to enable satisfactory morphological processing: stroke properties (e.g. length, thickness) measured directly on the binarised images would likely be inaccurate and heavily quantised. To mitigate this issue and enable sub-pixel accuracy in the measurements, we propose to use the following processing steps: 1. *upscale* (e.g. ×4, to 112×112[1]); 2. *binarise* (e.g. threshold ≥128); 3. compute *Euclidean distance transform* (EDT) from boundaries; 4. *skeletonise* (medial axis, i.e. ridges of EDT); 5. apply *perturbation* (cf. Section 3); and 6. *downscale* to original resolution.

We illustrate the pipeline in Fig. 2. The binary high-resolution digits have smooth boundaries and faithfully capture subtle variations in contour shape and stroke thickness that are only vaguely discernible in the low-resolution images. Additionally, note how the final downscaled image is almost indistinguishable from the original.

All morphometric attributes described below are calculated for each digit after applying steps 1–4 of this pipeline. The distributions for the plain MNIST training set is plotted in Fig. 3, and the distributions after applying each type of perturbation can be found in Appendix A.

Figure 3: Distribution of morphological attributes per digit class in the plain MNIST training dataset

---

[1]Up- and downscaling by a factor of $f$ are done with bicubic interpolation and Gaussian smoothing (bandwidth $\sigma = 2f/6$), following `scikit-image` defaults (van der Walt et al., 2014).

## 2.2 STROKE LENGTH

Here we approximate the trace of the pen tip, as a digit was being written, by the computed morphological skeleton. In this light, the total length of the skeleton is an estimate of the length of the pen stroke, which in turn is a measure of shape complexity.

It can be computed in a single pass by accumulating the Euclidean distance of each skeleton pixel to its immediate neighbours, taking care to only count the individual contributions once. This approach is more robust against rotations than a naïve estimate by simply counting the pixels.

## 2.3 STROKE THICKNESS

A prominent factor of style variation in the MNIST digits is the overall thickness of the strokes, due to both legitimate differences in pen thickness and force applied, and also to the rescaling of the original NIST images by different factors.

We estimate it by exploiting the computed distance transform. By virtue of how the image skeleton is computed, its pixels are approximately equidistant to the nearest boundaries, therefore we take twice the mean value of the EDT over all skeleton pixels as our global estimate.

## 2.4 SLANT

The extent by which handwritten symbols lean right or left (forward and backward slant, respectively) is a further notorious and quantifiable dimension of handwriting style. It introduces so much variation in the appearance of characters in images that it is common practice in OCR systems to 'deslant' them, in an attempt to reduce within-class variance (LeCun et al., 1998; Teow and Loe, 2002).

We adapt the referred deslanting methodology to describe the slant *angle* of the handwritten digits. After estimating the second-order image moments, we define the slant based on the horizontal shear:

$$\alpha = \arctan\left(-\frac{\sum_{i,j} x_{ij}(i - \bar{i})(j - \bar{j})}{\sum_{i,j} x_{ij}(i - \bar{i})^2}\right) , \tag{1}$$

where $x_{ij}$ is the intensity of pixel $(i, j)$, and $(\bar{i}, \bar{j})$ are the centroid coordinates. The minus sign ensures that positive and negative values correspond to forward and backward slant, respectively.

## 2.5 WIDTH AND HEIGHT

It is useful to measure other general shape attributes, such as width, height, and aspect ratio, which also present substantial variation related to personal handwriting style.[2] To this end, we propose to fit a *bounding parallelogram* to each digit, with horizontal and slanted sides (cf. Fig. 1).

We sweep the image top-to-bottom with a horizontal boundary to compute a vertical marginal cumulative distribution function (CDF), and likewise left-to-right with a *slanted* boundary for a horizontal marginal CDF, with angle $\alpha$ as computed above. The bounds are then chosen based on equal-tailed intervals containing a given proportion of the image mass—98% in both directions (1% from each side) proved accurate and robust in our experiments.

## 3 PERTURBATIONS

As discussed in Section 1, we bring forward a number of morphological perturbations for MNIST digits, to enable interesting applications and experimentation. In this section, we detail these parametrisable transformations, categorised as global or local.

---

[2]Note that little variation in height is expected, since the original handwritten digits were scaled to fit a 20×20 box (LeCun et al., 1998). Nevertheless, a minority of digits were originally wider than they were tall, which explains the long tails in the distribution of heights (Fig. 3).

### 3.1 Global: Thinning and Thickening

The first pair of transformations we present is based on simple morphological operations: the binarised image of a digit is dilated or eroded with a circular structuring element. Its radius is set proportionally to the estimated stroke thickness (Section 2.3), so that the overall thickness of each digit will decrease or increase by an approximately fixed factor (here, -70% and +100%; see Figs. B.1 and B.2).

Since there is substantial thickness variability in the original MNIST data (cf. Fig. 3) and most thinned and thickened digits look very plausible, we believe that these perturbations can constitute a powerful form of data augmentation for training. For the same reason, we have not included these perturbations in the abnormality detection experiments (Appendix D).

### 3.2 Local: Swelling

In addition to the global transformations above, we introduce *local* perturbations with variable location and extent, which are harder to detect automatically. Given a radius $R$, a centre location $\mathbf{r}_0$ and a strength parameter $\gamma > 1$, the coordinates $\mathbf{r}$ of pixels within distance $R$ of $\mathbf{r}_0$ are nonlinearly warped according to a radial power transform:

$$\mathbf{r} \mapsto \mathbf{r}_0 + (\mathbf{r} - \mathbf{r}_0)\Big(\frac{\|\mathbf{r} - \mathbf{r}_0\|}{R}\Big)^{\gamma-1}, \tag{2}$$

leaving the remaining portions of the image untouched and resampling with bicubic interpolation.

In the experiments and released dataset, we set $\gamma = 7$ and $R = 3\sqrt{\theta}/2$, where $\theta$ is thickness. Unlike simple linear scaling with $\theta$, this choice for $R$ produces noticeable but not exaggerated effects across the thickness range observed in the dataset (cf. Fig. B.3). The centre location, $\mathbf{r}_0$, is picked uniformly at random from the pixels along the estimated skeleton.

### 3.3 Local: Fractures

We describe the proposed procedure for adding fractures to an MNIST digit, where we define a fracture as a break in the continuity of a pen stroke. Because single fractures can in many cases be easily mistaken for true gaps between strokes, we add multiple fractures to each affected digit.

When selecting the location for a fracture, we attempt to avoid getting too close to stroke tips (points on the skeleton with a single neighbour) or fork points (more than two neighbours). This is achieved by sampling only among those skeleton pixels above a certain distance to these detected points. In addition, we would like fractures to be transversal to the pen strokes. Local orientation is determined based on second-order moments of the skeleton inside a window centred at the chosen location, and the length of the fracture is estimated from the boundary EDT. Finally, the fracture is drawn onto the high-resolution binary image with a circular brush along the estimated normal.

In practice, we found that adding three fractures with $1.5\,\mathrm{px}$ thickness, $2\,\mathrm{px}$ minimum distance to tips and forks and angle window of $5{\times}5\,\mathrm{px}^2$ ('px' as measured in the low resolution image) produces detectable but not too obvious perturbations (see Fig. B.4). We also extend the lines on both ends by $0.5\,\mathrm{px}$ to add some tolerance.

## 4 Evaluation Case Studies

In this section, we demonstrate potential uses of the proposed framework: using morphometrics to characterise the distribution of samples from generative models and finding associations between learned latent representations and morphometric attributes. In addition, we exemplify in Appendix D a variety of supervised tasks on the MNIST dataset augmented with perturbations.

### 4.1 Sample Diversity

Here we aim to illustrate ways in which the proposed MNIST morphometrics may be used to visualise distributions learned by generative models and to quantify their agreement with the true data distribution in terms of these semantic attributes. We also believe that extracting such measurements from model samples is a step toward diagnosing the issue of mode collapse.

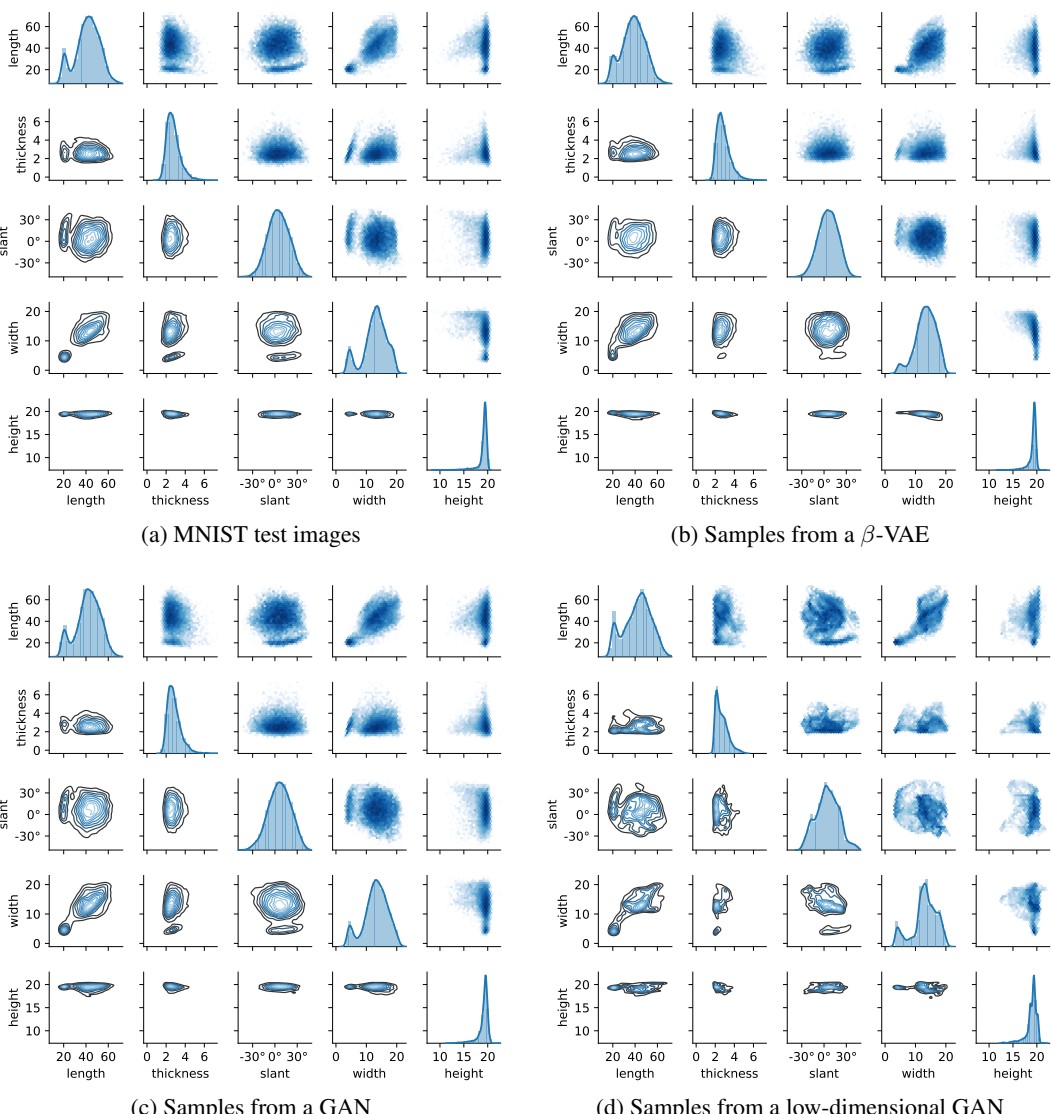

Figure 4: Distribution of morphometric attributes for MNIST test dataset and samples from some generative models. Diagonals show marginal histograms and KDEs, upper-triangular plots show pairwise log-histograms and lower-triangular plots show pairwise KDEs.

We exemplify this scenario with a vanilla GAN (Goodfellow et al., 2014) and a $\beta$-VAE (Higgins et al., 2017), both with generator (resp. decoder) and discriminator architecture as used in the MNIST experiments in Chen et al. (2016), and encoder mirroring the decoder. We train a $\beta$-VAE with $\beta = 4$ and a GAN, both with 64-dimensional latent space. To explore the behaviour of a much less expressive model, we additionally train a GAN with only two latent dimensions.

**Visualisation:** Figure 4 illustrates the morphometric distributions of the plain MNIST test images and of 10,000 samples from each of these three models. As can be seen, morphometrics provide *interpretable* low-dimensional statistics which allow comparing distributions learned by generative models between each other and with true datasets. While Figs. 4b and 4c show model samples roughly as diverse as the true images, the samples from the low-dimensional GAN in Fig. 4d seem concentrated on certain regions, covering a distribution that is less faithful to the true one in Fig. 4a.

**Statistical comparison:** We argue that in this lower-dimensional space of morphometrics it is possible to statistically compare the distributions, since this was shown not to be effective directly in

Table 1: Kernel two-sample tests between model samples and true test data

| Test data vs. | Dims. | $\text{MMD}_l^2 \pm$ std. error ($\times 10^{-3}$) | $p$ |
|---|---|---|---|
| $\beta$-VAE | 64 | $0.792 \pm 1.569$ | .3068 |
| GAN | 64 | $1.458 \pm 1.650$ | .1885 |
| GAN | 2 | $\mathbf{8.876 \pm 1.807}$ | **.0000** |

Table 2: Settings for InfoGAN disentanglement experiments

| | # Cat. | # Cont. | # Bin. | Dataset | |
|---|---|---|---|---|---|
| INFOGAN-A | 10 | 2 | 0 | PLAIN: | plain only |
| INFOGAN-B | 10 | 3 | 0 | GLOBAL: | plain + thinning + thickening |
| INFOGAN-C | 10 | 2 | 2 | LOCAL: | plain + swelling + fractures |

image space (e.g. Theis et al., 2016). To this end, we propose to use kernel two-sample tests based on maximum mean discrepancy (MMD) between morphometrics of the test data and of each of the sample distributions. Here, we performed the linear-time asymptotic test described in Gretton et al. (2012, §6) (details and further considerations in Appendix C). The test results in Table 1 seem to confirm the mismatch of the low-dimensional GAN's samples, whereas the $\beta$-VAE and larger GAN do not show a significant departure from the data distribution.

**Finding replicas:** One potentially fruitful suggestion would be to use a variant of hierarchical agglomerative clustering on sample morphometric attributes (e.g. using standardised Euclidean distance, or other suitable metrics). With a low enough distance threshold, it would be possible to identify groups of near-replicas, the abundance of which would signify mode collapse. Alternatively, this could be applicable as a heuristic in the birthday paradox test for estimating the support of the learned distribution (Arora et al., 2018).

## 4.2 DISENTANGLEMENT

In this experiment, we demonstrate that: (a) standard MNIST can be augmented with morphometric attributes to quantitatively study representations computed by an *inference* model (as already possible with e.g. dSprites and 3D faces); (b) we can measure shape attributes of samples to assess disentanglement of a *generative* model, which is unprecedented to the best of our knowledge; and (c) this analysis can also diagnose when a model unexpectedly *fails* to learn a known aspect of the data.

**Methodology:** We take MAP estimates of latent codes for each image (i.e. maximal logit for categorical codes and mean for continuous codes), as predicted by the variational recognition network. Using an approach related to the disentanglement measure introduced in Kumar et al. (2018), we study the correlation structures between known generative factors and latent codes learned by an InfoGAN. Specifically, we compute the *partial correlation* between each latent code variable and each morphometric attribute, controlling for the variation in the remaining latent variables (disregarding the noise vector).[3] As opposed to the simple correlation, this technique allows us to study the *net* first-order effect of each latent code, all else being equal.

Models were trained for 20 epochs using 64 images per batch, with no hyperparameter tuning. We emphasize that our goal was to illustrate how the proposed morphometrics can serve as tools to better understand whether they behave as intended and not to optimally train the models in each scenario.

**Inferential disentanglement:** To illustrate how this methodology can be applied in practice to assess disentanglement, we consider two settings. The first is the same as in the MNIST experiment from Chen et al. (2016), with a 10-way categorical and two continuous latent codes, trained and evaluated on the plain MNIST digits, which we will refer to as INFOGAN-A.

---

[3]For the categorical code, $c_1$, we take a single binary dummy variable for each category, $c_1^{(k)}$, while controlling only for the remaining codes ($c_2$, $c_3$ etc.) to avoid multicollinearity.

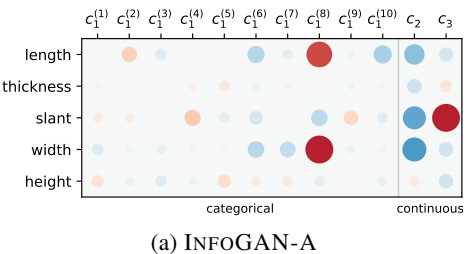

(a) INFOGAN-A

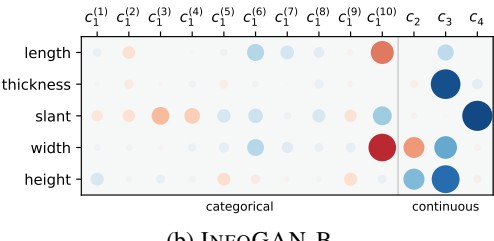

(b) INFOGAN-B

Figure 5: Partial correlations between inferred latent codes and morphometrics of test images. Circle area and colour strength are proportional to correlation magnitude, blue is positive and red is negative.

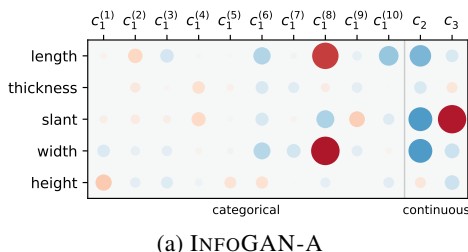

(a) INFOGAN-A

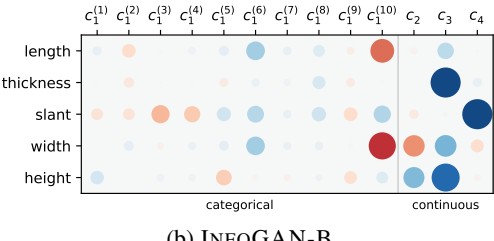

(b) INFOGAN-B

Figure 6: Partial correlations between 1000 sampled latent codes and morphometrics of the corresponding generated images

The second setting was designed to investigate whether the model could disentangle the concept of *thickness*, by including an additional continuous latent code and training on a dataset with exaggerated thickness variations. We constructed this dataset by randomly interleaving plain, thinned and thickened digit images in equal proportions. Since the perturbations were applied completely at random, we expect a trained generative model to identify that thickness should be largely independent of the other morphological attributes. We refer to this set-up as INFOGAN-B. Table 2 summarises the different experimental settings, for reference.

In Fig. 5a, we see that INFOGAN-A learned to encode slant mostly in $c_3$, while $c_1^{(8)}$ clearly relates to the '1' class (much narrower digit shape and shorter pen stroke; cf. Fig. 3). Figure 5b quantitatively confirms the hypothesis that INFOGAN-B's recognition network would learn to separate slant and thickness (in $c_4$ and $c_3$, resp.), the most prominent factors of style variation in this dataset. Interestingly, it shows that $c_3$ also associates with height, as thicker digits tend to be taller.

**Generative disentanglement:** The evaluation methodology described above is useful to investigate the behaviour of the *inference* direction of a model, and can readily be used with datasets which include ground-truth generative factor annotations. On the other hand, unless we trust that the inference approximation is highly accurate, this tells us little about the *generative* expressiveness of the model. This is where computed metrics truly show their potential: we can measure generated samples, and see how their attributes relate to the latent variables used to create them.

Figure 6 shows results for a similar analysis to Fig. 5, but now evaluated on *samples* from that model. As the tables are mostly indistinguishable, we may argue that in this case the inference and generator networks have learned to consistently encode and decode the digit shape attributes.

As further illustration, Fig. 7 displays traversals of the latent space, obtained by varying a subset of the latent variables while holding the remaining ones (including noise) constant. With these examples, we are able to qualitatively verify the quantitative results in Fig. 6. Note that, until now, visual inspection was typically the only means of evaluating disentanglement and expressiveness of the generative direction of image models (e.g. Chen et al., 2016; Dupont, 2018).

**Diagnosing failure:** We also attempted to detect whether an InfoGAN had learned to discover local perturbations (swelling and fractures). To this end, we extended the model formulation with additional Bernoulli latent codes, which would hopefully learn to encode presence/absence of each

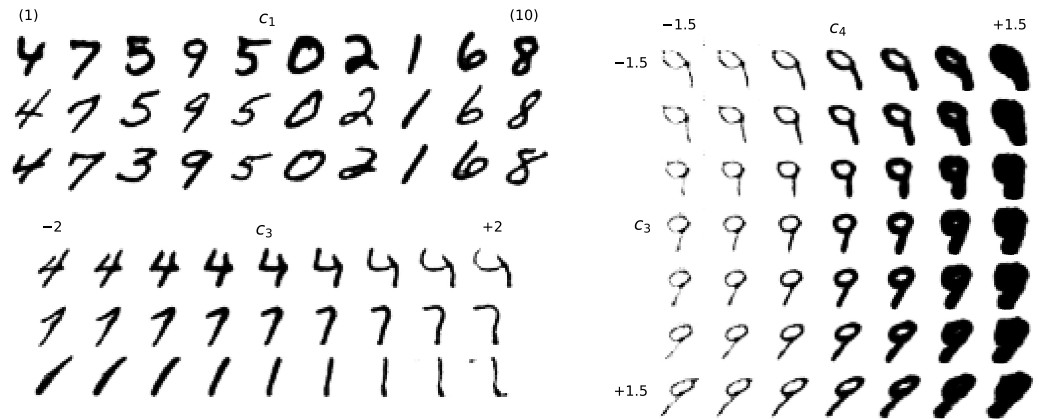

(a) INFOGAN-A: one-dimensional traversals of $c_1$ (*top*, 'digit type') and $c_3$ (*bottom*, 'slant'). Samples in each row share the values of remaining latent variables and noise.

(b) INFOGAN-B: two-dimensional traversal of $c_4 \times c_3$ ('thickness' $\times$ 'slant')

Figure 7: InfoGAN latent space traversals

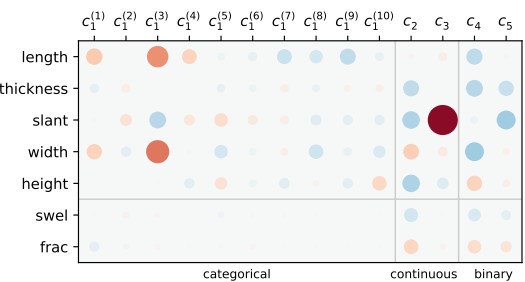

Figure 8: Partial correlations of inferred latent codes with test morphometrics (INFOGAN-C)

type of local perturbation. The model investigated here, dubbed INFOGAN-C (cf. Table 2), had a 10-way categorical, two continuous and two binary codes, and was trained with a dataset of plain, swollen and fractured digits (randomly mixed as above).

Again via inferential partial correlation analysis—now including ground-truth perturbation annotations—we can quantitatively verify that this particular model instance was unable to meaningfully capture the perturbations (Fig. 8, bottom-right block). In fact, it appears that the addition of the binary variables did not lead to more expressive representations in this case, even impairing the disentanglement of the categorical variables, if compared to Figs. 5a and 5b, for example.

## 5 CONCLUSION

With Morpho-MNIST we provide a number of mechanisms to quantitatively assess representation learning with respect to measurable factors of variation in the data. We believe that this is an important asset for future research on generative models, and we would like to emphasize that the proposed morphometrics can be used *post hoc* to evaluate already trained models, potentially revealing novel insights and interesting observations.

A similar morphometry approach could be used with other datasets such as dSprites, e.g. estimating shape location and size, number of objects/connected components. Perhaps some generic image metrics may be useful for analysis on other datasets, e.g. relating to sharpness or colour diversity, or we could even consider using the output of object detectors (analogously to the Inception-based scores; e.g. number/class of objects, bounding boxes etc.). In future work we plan to include additional perturbations, for example, mimicking imaging artefacts commonly observed in medical imaging modalities to add further complexity and realism.

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

# A    MORPHOMETRICS OF PLAIN AND PERTURBED DATASETS

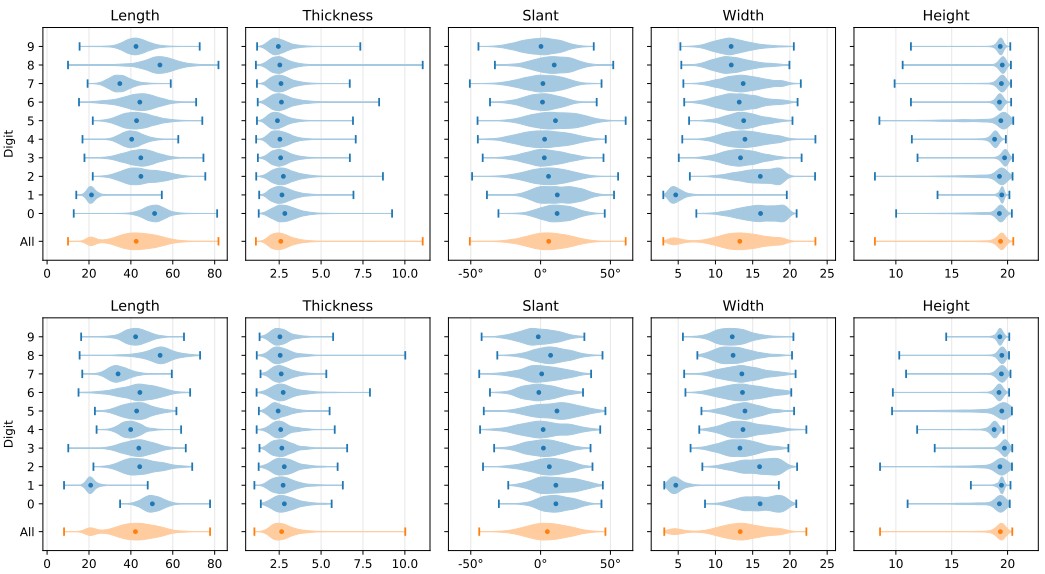

Figure A.1: Distribution of morphological attributes for plain MNIST digits. *Top:* training set; *bottom:* test set.

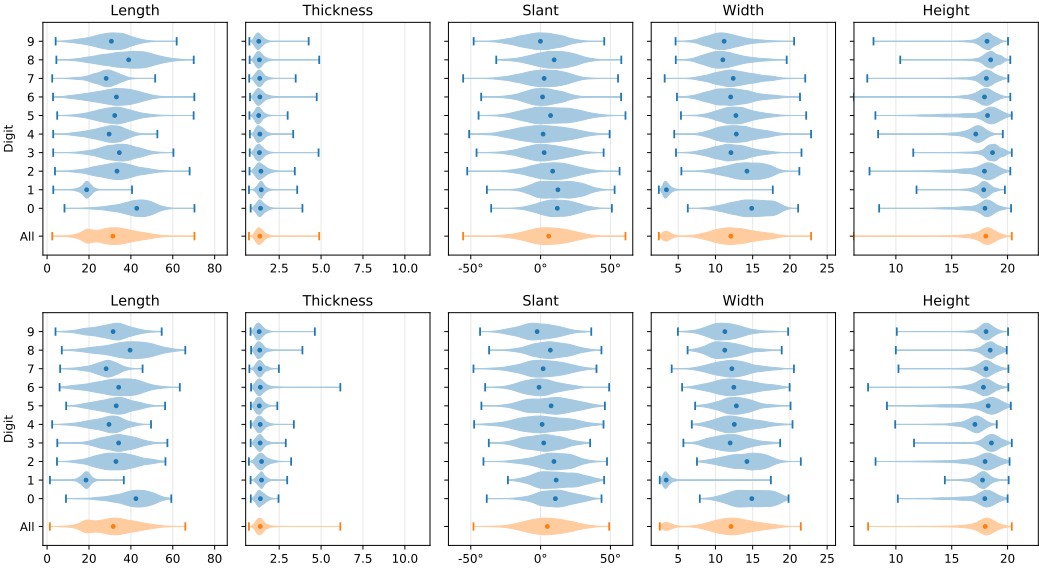

Figure A.2: Distribution of morphological attributes for thinned MNIST digits. *Top:* training set; *bottom:* test set.

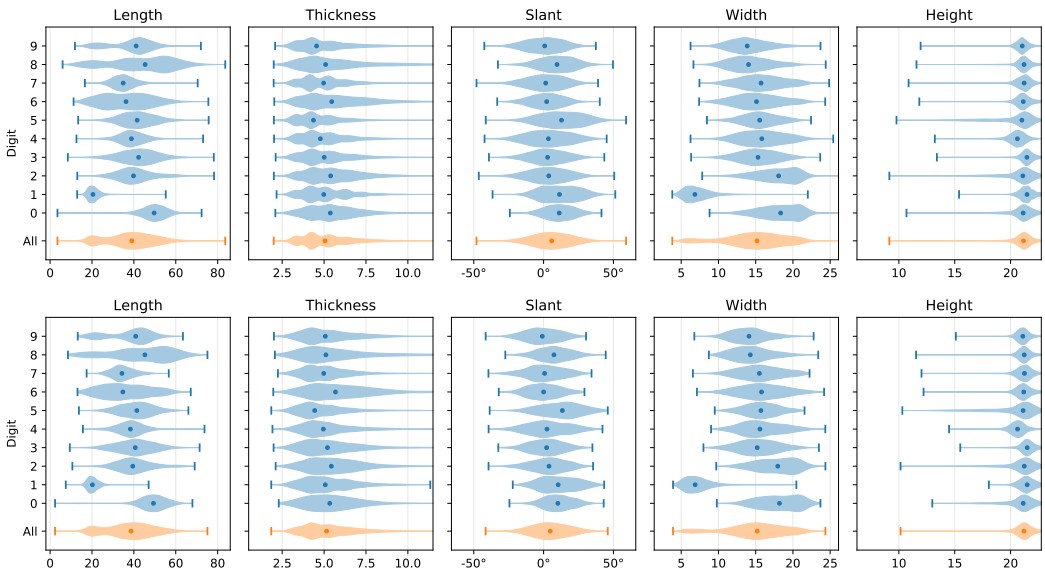

Figure A.3: Distribution of morphological attributes for thickened MNIST digits. *Top:* training set; *bottom:* test set.

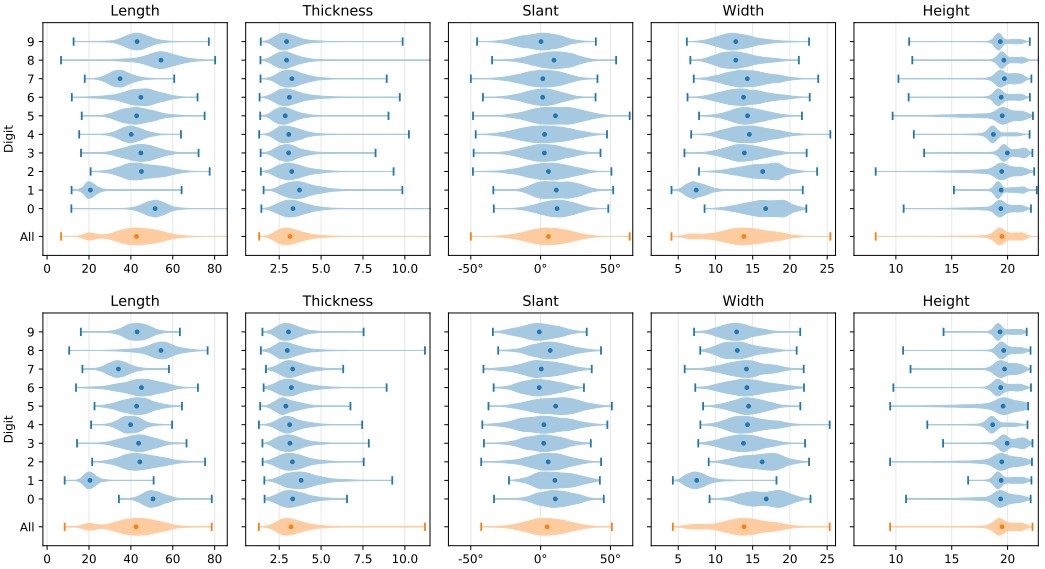

Figure A.4: Distribution of morphological attributes for swollen MNIST digits. *Top:* training set; *bottom:* test set.

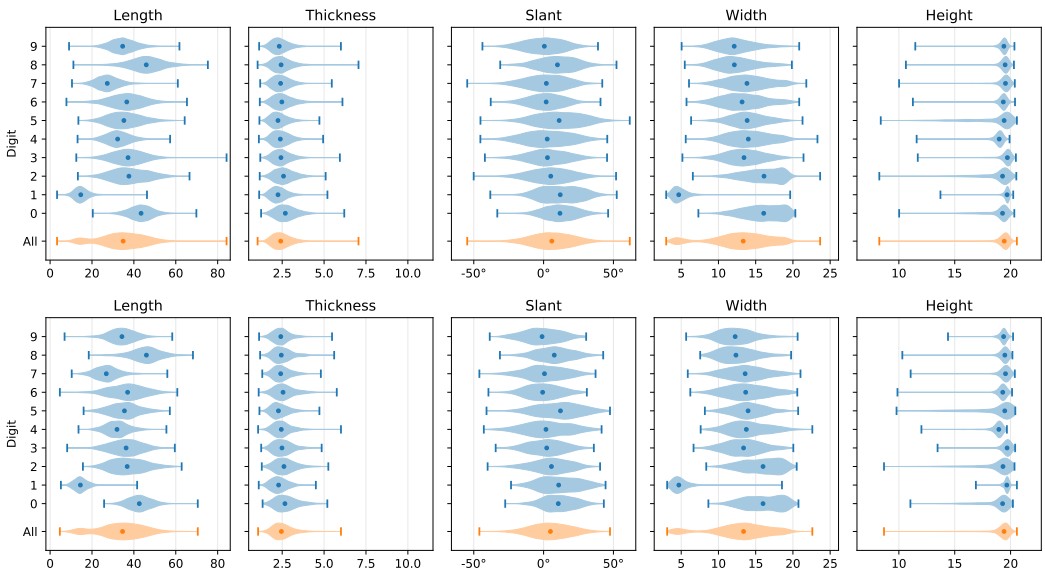

Figure A.5: Distribution of morphological attributes for fractured MNIST digits. *Top:* training set; *bottom:* test set.

## B PERTURBATION EXAMPLES

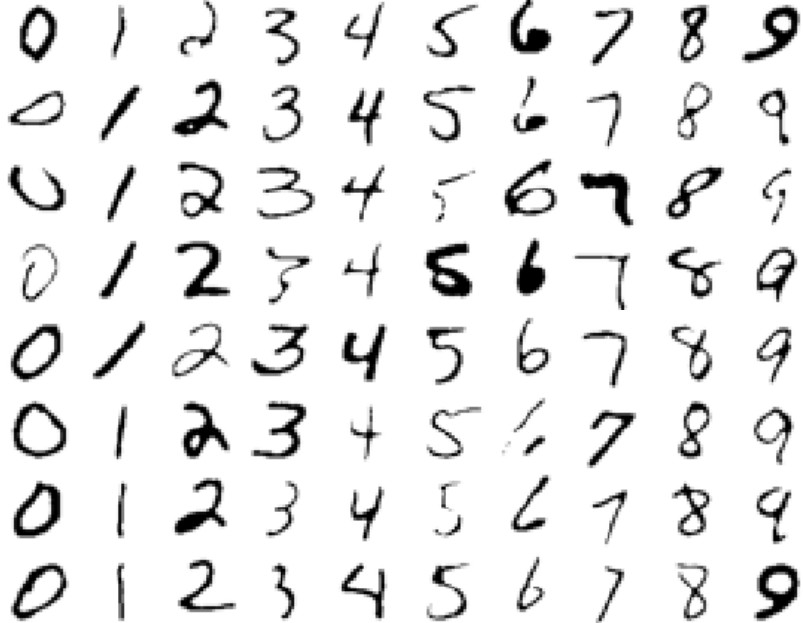

Figure B.1: Examples of globally thinned digits

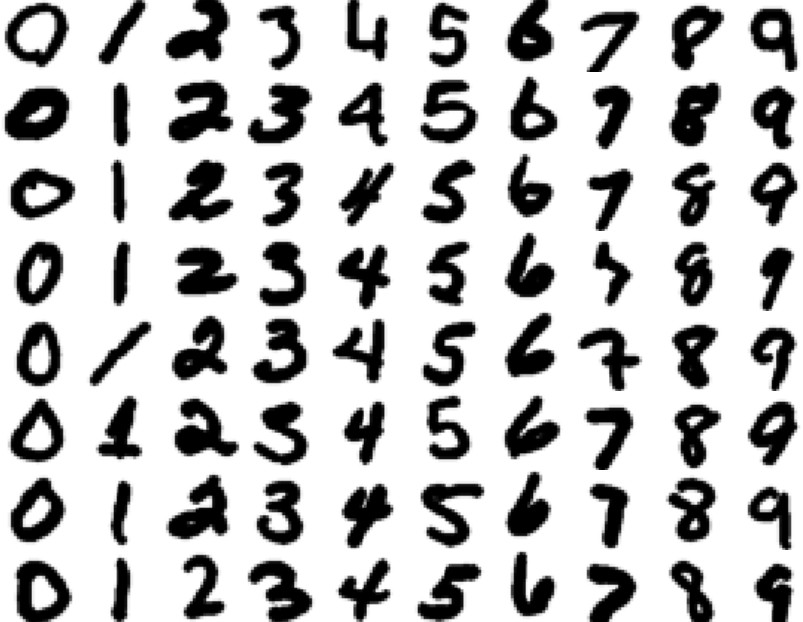

Figure B.2: Examples of globally thickened digits

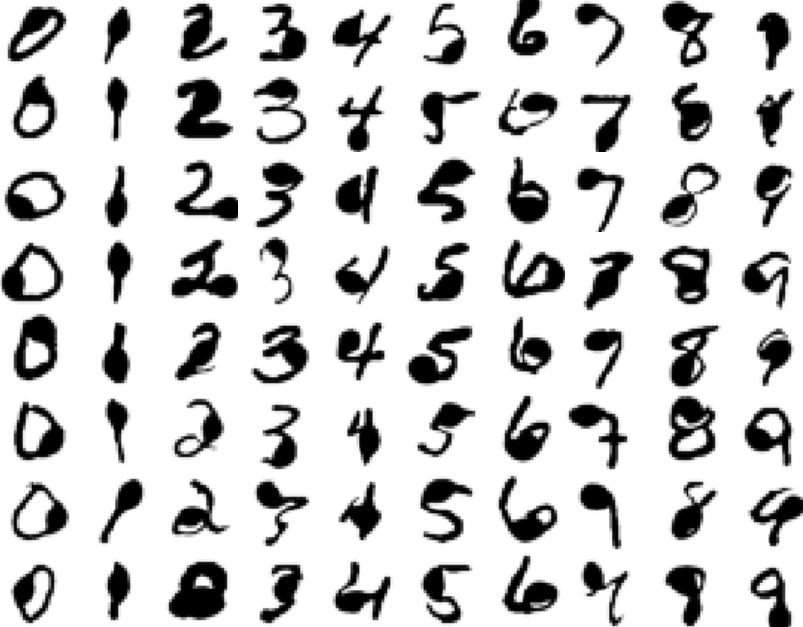

Figure B.3: Examples of digits with local swellings

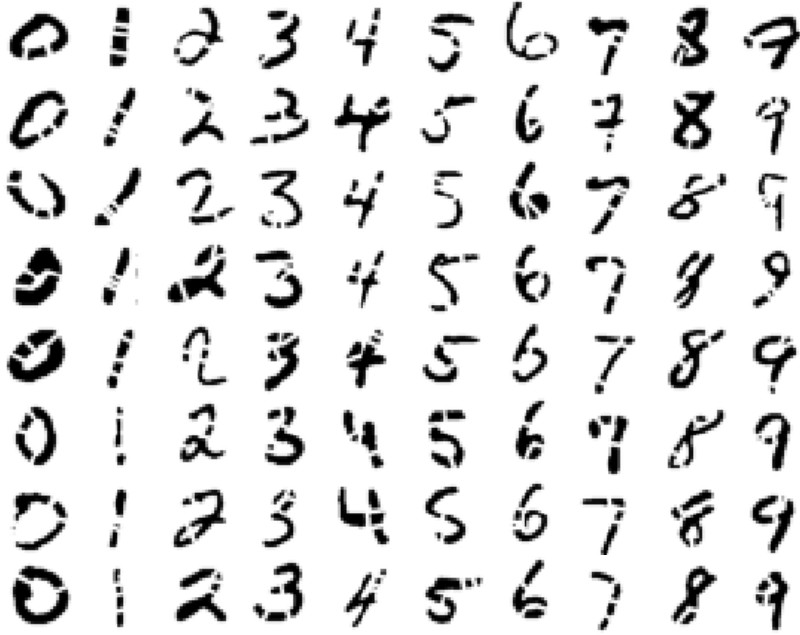

Figure B.4: Examples of digits with local fractures

## C  MMD DETAILS

We employed a Gaussian product kernel with bandwidths derived from Scott's rule, analogously to the KDE plots in Fig. 4. Scott's rule of thumb defines the bandwidth for a density estimation kernel as $N^{-1/(D+4)}$ times the standard deviation in each dimension, where $N$ and $D$ denote sample size and number of dimensions (Scott, 1992, Eq. (6.42)). We determine the KDE bandwidths separately for real and sample data, then add their squares to obtain the squared bandwidth of the MMD's Gaussian kernel, as it corresponds to the *convolution* of the density estimation kernels chosen for each set of data. See Gretton et al. (2012, §3.3.1) for further details on the relation between MMD and $L_2$ distance of kernel density estimates.

Whereas the bandwidth heuristic used here is fairly crude, much more sophisticated kernel selection procedures are available, e.g. by explicitly optimising the test power (Sutherland et al., 2017). A further analysis tool in a similar vein would be to apply a *relative* MMD similarity test (Bounliphone et al., 2016), to rank trained models based on sample fidelity. It would also be possible to adopt a model criticism methodology based on the MMD witness function (Lloyd and Ghahramani, 2015), to identify over- and under-represented regions in morphometric space (and corresponding generated image exemplars could be inspected as well).

## D  SUPERVISED TASKS

Although the driving motivation for introducing Morpho-MNIST has been the lack of means for quantitative evaluation of generative models, the proposed framework may also be a valuable resource in the context of supervised learning. We conducted several experiments to demonstrate potential applications of these datasets with increased difficulty due to the injected perturbations: standard digit recognition, supervised abnormality detection, and thickness regression. Note such experiments can later serve as baselines for unsupervised tasks such as outlier detection and domain adaptation.

We evaluated four different models: $k$-nearest-neighbours ($k$NN) using $k = 5$ neighbours and $\ell_1$ distance weighting, a support vector machine (SVM) with polynomial kernel and penalty parameter $C = 100$, a multi-layer perceptron (MLP) with 784–200–200–$L$ architecture ($L$: number of outputs), and a LeNet-5 convolutional neural network (LeCun et al., 1998). Here, we use the same datasets as in the disentanglement experiments (Section 4.2): plain digits (PLAIN), plain mixed with thinned and thickened digits (GLOBAL), and plain mixed with swollen and fractured digits (LOCAL).

For digit recognition, each model is trained once on PLAIN, then tested on both PLAIN and LOCAL test datasets, to investigate the effect of *domain shift*. All methods suffer a drop in test accuracy on LOCAL (Table 3, first two columns). $k$NN appears to be the most robust to the local perturbations, perhaps because they affect only a few pixels, leaving the image distance between neighbours largely unchanged. On the other hand, local patterns that LeNet-5 relies on may have changed considerably.

The abnormality detection task is, using the LOCAL dataset, to predict whether a digit is normal or perturbed (swollen or fractured)—compare with lesion detection in medical scans. Table 3 (third column) indicates that LeNet-5 is able to detect abnormalities with high accuracy, likely thanks to local invariances of its convolutional architecture. Note that all scores (especially the simpler models') are lower than digit classification accuracy, revealing the (possibly surprising) higher difficulty of this binary classification problem compared to the ten-class digit classification.

Finally, we also constructed a regression task for digit thickness using the GLOBAL dataset, mimicking medical imaging tasks such as estimating brain age from cortical grey matter maps. Since this is a non-trivial task, requiring some awareness of local geometry, it is perhaps unsurprising that the convolutional model outperformed the others, which rely on holistic features (Table 3, last column).

Table 3: Accuracy on supervised tasks using the proposed data perturbations

| Model | Digit Recognition (%) | | Abnormality Detection (%) | Thickness Regression (RMSE, pixels) |
|---|---|---|---|---|
| | PLAIN | LOCAL | | |
| $k$NN | 96.25 | 95.22 | 65.10 | 0.4674 |
| SVM | 95.71 | 92.47 | 77.59 | 0.3647 |
| MLP | 97.97 | 93.15 | 88.25 | 0.3481 |
| LeNet-5 | 98.95 | 95.33 | 97.53 | 0.2790 |

