# OpenReview forum: "Morpho-MNIST: Quantitative Assessment and Diagnostics for Representation Learning"
_ICLR.cc/2019/Conference_

### Official Review · AnonReviewer3 · 2018-11-03
**not enough contribution**

**Rating:** 4
**Confidence:** 3

**Review:**

The author proposed an extended version of MNIS where they introduced thickening/thinning/swelling/fracture. The operation is done using binary morphological operations.

* Providing benchmark data for tasks such disentanglement is important but I am not sure generating data is sufficient contribution for a paper.
* I am not sure what conclusion I should draw from Fig 5 and Fig 6 about the data.
* Eventually this data can become a benchmark data when it is paired with a method. Then that method/data are a benchmark.

---

> ### Author Response · Authors · 2018-11-07
> **Inadequate review, main contributions ignored**
>
> With all due respect, the reviewer’s summary of our work is inaccurate, incomplete and ignores the key part of our contributions. This suggests that the reviewer may not have carefully read our paper. The review consists of three lines, which we will address below. We strongly believe that the overall recommendation based on the reviewer’s comments is unjustified.
>
> Regarding the first statement “I am not sure generating data is sufficient contribution for a paper”:
>
> Generating a dataset is a relatively minor portion of this work; please refer to Sec. 1.1 for a clear outline of the main points. The bulk of our paper is about how morphometrics enable quantitative evaluation of learned representations (Section 4), and how the proposed image perturbations can enrich this evaluation and also open up a variety of new supervised tasks (explored in Appendix D).
>
> Regarding the second statement “I am not sure what conclusion I should draw from Fig 5 and Fig 6 about the data”:
>
> As described clearly in the text, Fig. 5 shows quantitative results for _inferential_ disentanglement (i.e. from real MNIST test data to latent codes) of two different InfoGANs, and Fig. 6 for _generative_ disentanglement (from latent codes to generated samples). In the paper we state “as the tables are mostly indistinguishable, we may argue that in this case the inference and generator networks have learned to consistently encode and decode the digit shape attributes.” To the best of our knowledge, this is the first time that partial correlations have been used to illustrate and quantitatively characterise the performance of representation learning, thanks to extracted morphometric attributes as proposed in the paper.
>
> The final statement “Eventually this data can become a benchmark data when it is paired with a method. Then that method/data are a benchmark” is not very clear to us. Our paper introduces both a novel quantitative assessment _methodology_ and new _datasets_ for experimentation and benchmarking of representation learning methods. We provide baseline results for recent approaches, including different variants of GANs and VAEs.
>
> In this light, we would like to reiterate that we strongly believe the reviewer’s assessment of our paper is inadequate.

---

### Official Review · AnonReviewer2 · 2018-11-03
**Interesting characterisation and extension of MNIST**

**Rating:** 5
**Confidence:** 3

**Review:**

Authors present a set of criteria to categorize MNISt digists (e.g. slant, stroke length, ..) and a set of interesting perturbations (swelling, fractures, ...) to modify MNIST dataset. They suggest analysing performance of generative models based on these tools. By extracting this kind of features, they effectively decrease the dimmension of  data. Therefore, statistically comparing the distribution of generated vs test data and binning the generated data is now possible. They perform a thorough study regarding MNIST. Their tools are a handy addition to the analytical surveys in several applications (e.g. how classification fails), but not convincingly for generation.

Since their method is manually designed for MNIST, the manuscript would benefit from a justification or discussion on the  common pitfalls and the correlation between MNIST generation and more complex natural image generation tasks. Since the presented metrics do not show a significant difference between the VAE and Vanilla GAN model, the question remains whether evaluating on MNIST is a good proxy for the performance of the model on colored images with backgrounds or not. For example sharpness and attending to details is not typically a challenge in MNIST generation where in other datasets this is usually the first challenge to be addressed. I'm not convinced that ability of a model in disentangling thickness correlates to their ability in natural image generation.

---

> ### Author Response · Authors · 2018-11-07
> **Representation learning vs. sample quality**
>
> We appreciate the thoughtful review and suggestions for adding clarifications, in particular with respect to the aspect of natural image generation.
>
> We completely agree that MNIST generation is not to be mistaken as a surrogate for the generation of natural images, and nowhere in our paper did we intend to suggest otherwise. We thank the reviewer for bringing up this potential confusion, and we will add further clarification to make sure there is no ambiguity about this in our paper.
>
> As a matter of fact, we believe that the current focus of research towards generating natural images can be misleading (or at least gives an incomplete picture) in the context of representation learning, as the quality of sampled images generally tells us little about how well the learned representations capture the known factors of variation in the training distribution.
>
> With Morpho-MNIST, we aimed to address this issue by providing an objective methodology for evaluating representation learning, i.e. quantitatively measuring how expressive a trained generative model is and how well it covers the variability in the data, in our case defined by morphometry of shapes represented as grayscale images. In fact, we make no statements about measuring sample _quality_, only the _diversity_ of shape attributes. Our conclusion does point to possible extensions of this framework to other measurable content attributes for different types of images.
>
> The fact that VAE and GAN performed similarly under our metrics shows only that, for this data and similar model capacities, the representations they learned are comparably expressive. We actually believe this is an important message to convey, as a large body of work focusing on the crispness of generated images might incorrectly lead to a conclusion that VAEs are generally inferior to GANs with respect to representation learning. Here, we can demonstrate quantitatively that for the considered type of distribution (morphometry of rasterised shapes) this is not the case.
>
> On your point of “whether evaluating on MNIST is a good proxy for the performance of the model on colored images with backgrounds or not” we would say the answer is clearly no. But, as mentioned above and hopefully made more clear in our revision, it was never our intention to imply otherwise. We also couldn’t agree more with your statement “I'm not convinced that [the] ability of a model in disentangling thickness correlates to their ability in natural image generation.” These are very different problems.
>
> However, we would like to reiterate some of our reasons for focusing on MNIST, as presented in the introduction: few and simple factors of variation, sufficient size for its complexity, low computational requirements, and availability. Importantly, MNIST is a standard baseline on which a great number of generative models proposed in the literature are evaluated. This means that our framework can also be applied to these models retrospectively, adding novel insights about their performance in a more objective and quantitative manner. We do believe this is an important contribution to the area of representation learning.
>
> Here are a few such prominent works, in addition to the ones cited in our paper:
> - Goodfellow et al. (NIPS 2014). Generative Adversarial Nets.
> - Nowozin et al. (NIPS 2016). f-GAN: Training Generative Neural Samplers using Variational Divergence Minimization.
> - Radford et al. (ICLR 2016). Unsupervised Representation Learning with Deep Convolutional Generative Adversarial Networks.
> - Salimans et al. (NIPS 2016). Improved Techniques for Training GANs.
> - Rezende & Mohamed (ICML 2015). Variational Inference with Normalizing Flows.
> - Mescheder et al. (ICML 2017). Adversarial Variational Bayes: Unifying Variational Autoencoders and Generative Adversarial Networks.

---

### Official Review · AnonReviewer1 · 2018-11-08
**Review: Morpho-MNIST: Quantitative Assessment and Diagnostics for Representation Learning**

**Rating:** 3
**Confidence:** 4

**Review:**

This paper discusses the problem of evaluating and diagnosing the representations learnt using a generative model. This is a very important and necessary problem.

However, this paper lacks in terms of experimental evaluation and has some technical flaws.
1. Morphological properties deals with only the "shape" properties of the image object. However, when the entire image is subject to the generative model, it learns multiple properties from the image apart from shape too - such as texture and color. Additionally, there are lot of low level pixel relations that the model learns to fit the distribution of the given images. However, here the authors have assumed that the latent space of the generative models are influenced only by the morphological properties of the image - which is wrong. Latent space features could be affected by the color or texture of the image as well.

2. Extracting morphological properties of the image is straight-foward for MNIST kind of objects. However, it becomes really difficult for other datasets such as CIFAR or some real world images. Studying the properties of a generative model on such datasets is very challenging and the authors have not added a discussion around that.

3. Now assuming that my GAN model has learnt good representation in Morpho-MNIST dataset, is it guaranteed to learn good representations in other datasets as well? There is no guarantee on generalizability or extensibility of the work.

---

> ### Author Response · Authors · 2018-11-09
> **Incorrect assumptions**
>
> We thank the reviewer for acknowledging the importance of the problem we aimed to address, however, we very much disagree with the statements made regarding our assumptions.
>
> Regarding the reviewer’s first point, we believe there is a misunderstanding. We absolutely agree that a generative model needs to learn about colour, texture, and low-level pixel relations to be able to extract its representations and to produce reasonable samples. Regarding the reviewer’s statement that “the authors have assumed that the latent space of the generative models are influenced only by the morphological properties of the image”, we would like to stress that we never made such assumptions nor have we claimed that the latent space of models trained on MNIST capture exclusively shape variations. What the Morpho-MNIST methodology aims to answer is: “to what extent has my model learned to represent these specific factors of variation in the data?” If colour and texture are important factors for a given application or dataset, it suffices to design the relevant scalar metrics and include them in the very same framework.
>
> This brings us to the second point. As far as we are aware, this is the first attempt in *any* context to quantitatively characterise inferential and generative behaviour of learned representations. We propose to do it in terms of measurable features: here we exploit shape attributes, and in the conclusion we point to various possible extensions involving colours or object properties. In our view, it just makes sense that the first step in that direction builds on a simple dataset with well understood and easily measurable factors of variation.
>
> Finally, although it is correct that there are no generalisability guarantees, that is the case for any model evaluated on MNIST, CIFAR-10, or even ImageNet (cf. https://arxiv.org/abs/1806.00451, for example). As argued above, we are proposing a toolset to inspect and diagnose trained generative models that works with any collection of measurable attributes. Evidently conclusions may not be transferable if the datasets have different relevant attributes.

---

### Author Response · Authors · 2018-11-17
**asking for reviewer feedback**

Dear Reviewers,

We appreciate your time on assessing our paper, and we would highly value to hear back whether our responses to the criticism and our additional argumentation may change your recommendation. We believe that our work could be of great interest to the ICLR community and may initiate a very much needed discussion about objective and quantitative evaluation in representation learning.

As we believe most of the main criticism was based on misunderstanding that can be easily addressed in an updated version of the paper, we would love to hear what you think. Is there anything you believe we missed in our responses, or any other points you would like us to address in order to improve our paper?

Many thanks again for your time and valuable feedback.

---

> ### Author Response · Authors · 2018-11-22
> **Revision uploaded with clarifications**
>
> Dear Reviewers,
>
> We have uploaded a revision of our paper, taking into account your feedback and attempting to clarify any misunderstandings, as outlined in our responses below.
>
> Please consider reevaluating the new version, and thank you once again.

---

### Meta-Review · Area_Chair1 · 2018-12-07
**a useful dataset, but not enough of a contribution for an ICLR paper**

**Confidence:** 4
**Recommendation:** Reject

**Metareview:**

This paper presents a dataset for measuring disentanglement in learned representations. It consists of MNIST digits, sometimes transformed in various ways, and labeled with a variety of attributes. This dataset is used to measure statistics of various learned models.

Measuring disentanglement is certainly an important problem in our field. This dataset seems to be well designed, and I would recommend its use for papers studying disentanglement. The experiments are well-designed. While the reviewers seem bothered by the fact that it's limited to MNIST, this doesn't strike me as a problem. We continue to learn a lot from MNIST, even today.

But producing a useful dataset isn't by itself a significant enough research contribution for an ICLR paper. I'd recommend publication if (a) it were very different from currently existing datasets, (b) constructing it required overcoming significant technical obstacles, or (c) the dataset led to particularly interesting findings.

Regarding (a), there are already datasets of similar complexity which have ground-truth attributes useful for measuring disentanglement, such as dSprites and 3D Faces. Regarding (b), the construction seems technically straightforward. Regarding (c), the experimental findings are plausible and consistent with past findings (which is a good validation of the dataset) but not obviously interesting in their own right.

So overall, this seems like a useful dataset, but I cannot recommend publication at ICLR.